# Comienzo Saludable Puerto Rico: A Community-Based Network of Care to Improve Maternal, Newborn, and Child Health Outcomes

**DOI:** 10.3390/ijerph22081204

**Published:** 2025-07-31

**Authors:** Edna Acosta-Pérez, Cristina Díaz, Atisha Gómez-Reyes, Samaris Vega, Carlamarie Noboa Ramos, Rosario Justinianes-Pérez, Glamarie Ferran, Jessica Carnivali-García, Fabiola J. Grau, Lili M. Sardiñas, Maribel Campos, Marizaida Sánchez Cesareo

**Affiliations:** 1Grupo Nexos, Inc., Guaynabo 00966-2715, Puerto Rico; agomez@nexospr.org (A.G.-R.); svega@nexospr.org (S.V.); cnoboa@nexospr.org (C.N.R.); rjustinianes@nexospr.org (R.J.-P.); jcarnivali@nexospr.org (J.C.-G.); fgrau@nexospr.org (F.J.G.); lsardinas@nexospr.org (L.M.S.); msanchez@nexospr.org (M.S.C.); 2Urban Strategies, LLC, Arlington, VA 22209, USA; cdiaz@urbanstrategies.us (C.D.); gferran@urbanstrategies.us (G.F.); 3Emerge PBC, Guaynabo 00966-2715, Puerto Rico; maribel@emergemos.org

**Keywords:** maternal health, newborn/child health, networks of care, community-based care, Puerto Rico, health differences, integrated care, infant mortality

## Abstract

Background: Maternal and newborn health disparities remain a challenge in Puerto Rico, especially in underserved communities. *Comienzo Saludable* Puerto Rico, sponsored by the U.S. Department of Health and Human Services’ Healthy Start Initiative (HRSA), addresses these gaps through an integrated Networks of Care model known as *Cuidado Compartido. Comienzo Saludable* Puerto Rico is a maternal, paternal, and child health program aimed at improving the health and well-being of pregnant women, mothers, fathers, newborns, and children in Puerto Rico, particularly those from disadvantaged communities. Methods: This paper presents the *Comienzo Saludable* Puerto Rico program’s *Cuidado Compartido* model to integrate a network of healthcare providers and services across hospitals, community organizations, and families. This model aims to improve maternal and newborn/child health outcomes by focusing on the importance of integrated, hospital-community-based care networks. Results: Participants experienced significant improvements in key birth outcomes: low birth weight prevalence declined by 27.2% compared to the community baseline, premature birth rates decreased by 30.9%, and infant mortality dropped by 75%, reaching 0% by 2021 and remaining there through 2023. These results were complemented by increases in maternal mental health screening, paternal involvement, and breastfeeding practices. Conclusions: The *Cuidado Compartido* model demonstrates a scalable, culturally responsive strategy to improve maternal, newborn, and child health outcomes. It offers critical insights for implementation in other high-need contexts.

## 1. Introduction

### 1.1. Maternal and Newborn/Child Health in Puerto Rico

Maternal and newborn/child health in Puerto Rico is marked by disproportionately high rates of preterm birth, low birth weight, and infant mortality, exacerbated by systemic issues related to healthcare access and quality [1]. Between 2000 and 2020, Puerto Rico experienced a population decline of over 830,000 people—a combined loss of 18%—including more than 15,000 healthcare professionals [2,3]. This outmigration significantly impacted the availability of clinical services, especially obstetric care. According to the Puerto Rico Department of Health, the number of practicing obstetricians has declined by 67% since 2010, with fewer than 150 OB/GYNs remaining by 2019. Many municipalities lack any obstetric coverage, contributing to the emergence of “maternity care deserts.” Maternity care deserts are defined—according to the March of Dimes—as counties where there is no hospital or birth center offering obstetric care and no obstetric clinicians (including obstetricians, certified nurse midwives, or family physicians providing delivery care) [4].

The shrinking population has also coincided with a drop in annual births to approximately 20,000 and the closure of over ten delivery rooms across the archipelago, intensifying the perinatal health crisis [3,5]. These challenges are most severe in disadvantaged communities, where access to essential maternal and newborn services remains critically limited.

In Puerto Rico, 20% of municipalities are classified as maternity care deserts—compared to 32.6% across the U.S.—highlighting persistent disparities despite a lower national average [5]. According to the March of Dimes, maternity care deserts are areas with no hospital or birth center offering obstetric care and no obstetric clinicians (e.g., OB/GYNs, certified nurse midwives, or family physicians providing delivery services) [5]. In 2023, among 18,645 births registered, the cesarean rate was 50.5%, infant mortality was 7.4 per 1000 live births, preterm birth was 12.4%, and low birthweight was 11.0% [6]. Relative to the U.S. mainland, Puerto Rico shows a 23% higher rate of preterm birth, 35% higher low birth weight, 38% higher infant mortality, 67% more teen births, and 75% more unintended pregnancies—exceeding the U.S. average of 49% by 65%. Additionally, the cesarean section rate on the archipelago exceeds 46%, significantly above the national average. Given this context, the *Comienzo Saludable* Shared Care Model (*Modelo de Cuidado Compartido* in Spanish) was developed to address the complex factors impacting maternal and newborn/child health outcomes. By fostering stronger coordination among healthcare providers, families, and communities, the model aims to create fair, accessible, responsive, and sustainable systems of care.

### 1.2. Networks of Care Framework

The *Cuidado Compartido* model is guided by the Networks of Care (NOC) framework, which promotes integrated, person-centered, and community-connected strategies across the health system. This approach aims to reduce fragmentation by strengthening linkages among hospitals, clinics, and community-based providers, with an emphasis on continuity, responsiveness, and equitable access to maternal and newborn care [7]. See Figure 1.

*Comienzo Saludable* Puerto Rico embodies this framework through its *Cuidado Compartido* model, which prioritizes communication, coordination, and shared responsibility across the continuum of care. By establishing strong partnerships among healthcare institutions, healthcare providers, public health programs, and community-based workers—including doulas, family health promoters (home visitors), and peer counselors—the model creates a support network centered around the unique needs of each pregnant or postpartum individual. This approach is especially vital in underserved areas of Puerto Rico, where healthcare access is limited, and the outcomes are disproportionately poor.

An NOC offers a strategic approach to addressing these challenges by creating interconnected systems that emphasize continuous, patient-centered care across settings and providers. By linking hospitals, clinics, community organizations, and families, Networks of Care can improve communication, foster accountability, and enhance health outcomes. The *Cuidado Compartido* model, implemented through *Comienzo Saludable* Puerto Rico, operationalizes this framework through shared care pathways, evidence-based practices, and community engagement.

The Networks of Care (NOC) framework, as outlined by Kalaris et al. [7], emphasizes the need for integrated, person-centered, and community-based approaches to improve maternal and newborn/child health outcomes. The model promotes the deliberate and coordinated linkage of health services across various levels of the health system—particularly between hospitals, clinics, and community-based organizations—to ensure continuity, quality, and access to care [7]. By fostering collaboration among healthcare providers, mobilizing community resources, and engaging families, the NOC model helps address structural fragmentation and gaps that often hinder the effective delivery of maternal and newborn care.

Empirical evidence supports the effectiveness of NOC-aligned models in improving patient-level outcomes, including perinatal depression screening and depression service referral rates [8]. Furthermore, integrated community-based approaches have been associated with significant gains in maternal and parenting practices, such as breastfeeding initiation and sustained paternal involvement [9,10].

*Comienzo Saludable* Puerto Rico embodies this framework through its *Cuidado Compartido* model, which prioritizes communication, coordination, and shared responsibility across the continuum of care. The model establishes partnerships among healthcare institutions, providers, public health programs, and community-based workers—including doulas, family health promoters (home visitors), and peer counselors—to deliver integrated, person-centered support tailored to each pregnant or postpartum individual. As noted by Kalaris et al. [7], NOC models foster communication and trust that are critical to strengthening maternal–child health outcomes, including screening practices, service uptake, and caregiver confidence. This approach is especially vital in underserved areas of Puerto Rico, where healthcare access is limited and outcomes are disproportionately poor.

Capacity-building is a core element of NOC strategies, with evidence showing that training frontline staff—including community health workers and doulas—can strengthen care delivery and improve outcomes [11,12]. Additionally, culturally responsive and relational approaches within NOC models have been linked to improved psychosocial outcomes such as caregiver empowerment, emotional well-being, and future health planning [13]. The Cuidado Compartido model operationalizes these principles through shared care pathways, evidence-based practices, and sustained community engagement—demonstrating how NOC strategies can be adapted to meet the needs of structurally vulnerable populations in Puerto Rico. This paper contributes to the growing body of implementation science by highlighting a community-based Networks of Care model designed to address maternal and newborn health disparities in Puerto Rico.

### 1.3. Purpose

This paper examines the structure, strategies, and impact of *Comienzo Saludable* Puerto Rico, an implementation of the Networks of Care framework designed to improve maternal and child health outcomes in underserved communities. Specifically, we analyze trends in key health indicators—including infant mortality, preterm births, and low birth weight—among program participants from 2019 to 2023. We also explore thematic sentiments expressed by patients and providers through open-ended intake responses, focusing on perceptions of empowerment, care quality, and service coordination. By showcasing its application and impact, we aim to present an integrated model for addressing and improving maternal and newborn/children health outcomes in other at-risk regions. The Program: *Comienzo Saludable* Puerto Rico

*Comienzo Saludable* Puerto Rico launched services in 2019 following a collective model framework and thus establishing a consortium of scientific community partnerships between Urban Strategies, LLC., St. Louis, MI, USA, (US) (US is a social enterprise founded with the desire to improve the health, educational, social, and economic condition of the families served, believing that by connecting, resourcing, and equipping grassroots organizations, the lives of vulnerable children and families can be dramatically improved. US equips, resources, and connects faith- and community-based organizations that are engaged in community transformation to help families reach their fullest potential) and Grupo Nexos, Inc., San Juan, Puerto Rico, (GN) (GN is an applied public health institute whose vision is to serve as a catalytic agent to effectively facilitate the dissemination and implementation of scientific innovation in the field of prevention and health promotion. GN’s mission is to implement scientifically proven intervention strategies in the community, providing knowledge and expert advice in areas such as public policy, planning, organizational development, program development, design and evaluation, training, and informatics). This collaborative partnership is characterized by three main elements: a social investor, a scientific partner, and an implementer (Kania & Kramer [14,15]). The social investor is the agency, organization, or foundation that acts as a funder for a project, program, or initiative. In this program, the U.S. Department of Health and Human Services (HRSA) serves as the investor of Healthy Start, developed as a community-based program that reduces health gaps in maternal and infant health [16]. The implementer is the agency or organization that executes the project, program, or initiative. For this program, US shares this role with selected staff from GN, designing and managing the implementation process. US maintains responsibility for human resources, finance, executive-level operations of the program, and monitoring to ensure accountability and program outcomes. The third element is the scientific partner, which works hand in hand with both social investors and implementers in all phases of the program to increase the probability of success. GN serves as both the co-implementer and scientific partner. While US maintains operational responsibility for human resources, finance, and executive oversight, GN leads the scientific and evaluation components of the program. This includes protocol design, fidelity monitoring, evaluation planning, training, and technical assistance. GN also collaborates in the implementation process by providing content expertise, capacity-building, and guidance to ensure the application of evidence-based practices. This dual role allows GN to bridge scientific integrity with community-based execution, ensuring rigor and responsiveness throughout the program lifecycle.

### 1.4. Cuidado Compartio Model: Key Program Components

*Comienzo Saludable* Puerto Rico was designed to increase access, strengthen social and familial support, and create a supportive culture for pregnant and parenting families. Since its inception in 2019, *Comienzo Saludable* Puerto Rico has been actively delivering maternal and child health services in the central-south region of the archipelago, with reported impacts on the communities of Jayuya, Ciales, Ponce, Juana Díaz, Santa Isabel, Arroyo, Salinas, and Guayama (See Figure 2). By implementing evidence-based practices, programs, and support services, CS has aimed to improve women’s health, family health, and overall wellness and foster systemic change.

The comprehensive goals of *Comienzo Saludable* are to (1) enhance access to high-quality healthcare and services for women, infants/children, and families; (2) strengthen the health workforce through capacity-building initiatives; (3) cultivate healthy communities and ensure continuous, coordinated, and comprehensive services; and (4) promote and advance health access by fostering connections with relevant organizations. In May 2024, *Comienzo Saludable* Puerto Rico expanded its services to impact 1400 participants annually across 24 municipalities in Puerto Rico (representing 31% of counties). *Comienzo Saludable* services consist of case management and navigation, comprehensive prenatal/care, postpartum care and support, infant/child care and follow-up, and health education and outreach. Figure 3 presents details of these.

The model follows not only relational and structural pillars for integrated, continuous, evidence-based, and family-centered care but also a community involvement pillar focusing on community-engaged, continuous monitoring and quality improvement. Program staff (e.g., family health promoters, navigators, and community health workers or providers, CHWs) serve as vital connectors, ensuring that families are linked to the appropriate services, including medical care, mental health support, nutrition counseling, transportation, and community-based programs. These services are grounded in the principles of care coordination and family/person-centered support, helping participants understand their care plans, overcome logistical and structural challenges, and follow through on recommended interventions [17,18]. By fostering strong communication and collaboration among families, healthcare providers, and community organizations, the case management component plays a critical role in promoting continuity of care, addressing social determinants of health, and improving maternal and infant/child outcomes. A special emphasis is placed on the active participation of families and local stakeholders, acknowledging that health outcomes are influenced not only by clinical care but also by social, cultural, and community contexts [13]. To this end, the program prioritizes the meaningful involvement of families, community health workers, and local organizations to foster trusted relationships, promote shared decision-making, and support culturally responsive care practices that contribute to sustained improvements in maternal and infant/child health [19,20]. These efforts are supported by a comprehensive training and technical assistance framework, alongside a continuous process of monitoring, evaluation, and quality improvement.

Finally, the program also places a strong emphasis on data collection, monitoring health outcomes, and tracking improvements in maternal and child health. To guarantee measurable improvement, particularly around healthcare services and the health status of the targeted population, a continuous quality improvement (CQI) plan was developed and carefully implemented. The purpose of the CQI is to optimize program outcomes, identify and disseminate best practices, and test innovation in comprehensive services provision that can increase efficiency and enhance the effectiveness of programs. This structure ensures that professionals are well-prepared and consistently deliver relevant, evidence-based services aligned with expected outcomes in the short, medium, and long term.

## 2. Materials and Methods

This study employed a program evaluation design to assess the implementation and outcomes of the *Comienzo Saludable* Puerto Rico initiative. The evaluation incorporated both quantitative and qualitative methods, drawing on administrative data, participant surveys, clinical records, and structured interviews to measure changes in maternal, newborn, and family health indicators over time. The focus was on real-world outcomes within a community-based service delivery context.

### 2.1. Population and Setting

*Comienzo Saludable* Puerto Rico specifically targets high-risk pregnant women, particularly those with limited access to healthcare services. This includes women from low-income backgrounds, those living in rural or underserved areas, and individuals who face barriers such as lack of transportation, living in remote areas, geographic isolation, limited social support, and elevated rates of infant mortality, premature births, and maternal complication rates. The initiative focuses on reducing these gaps by improving access to high-quality healthcare, health education, and comprehensive support services.

During implementation, an updated need assessment allowed for commitment to a selected target area but also expansion to other areas of need. Several changes in rates in the geographical area after 2019 provided an opportunity to scale up services (currently, 24 municipalities are being impacted).

### 2.2. Data Collection and Analysis

Program outcomes are assessed using a combination of administrative data, clinical records, participant surveys, and case management tracking systems. Analyses focus on trends in infant mortality rates, preterm birth incidence, low birth weight prevalence, maternal mental health screenings, and healthcare engagement rates. Tools are administered based on participant characteristics (e.g., pregnancy or postpartum status), clinical relevance, and the timing of service entry. Additional assessments may be completed over time depending on the care needs identified by health promoters.

#### 2.2.1. Participant Intake and Enrollment

*Comienzo Saludable* uses a centralized intake and service planning protocol to identify, assess, and enroll participants (quantitative and qualitative assessment items). Individuals who meet eligibility criteria (place of living) are scheduled for an initial intake appointment with a care coordinator/family health promoter. During this appointment, participants undergo a comprehensive assessment that includes domains such as health insurance coverage, access to medical care, health history, prenatal and postnatal care, reproductive life planning, mental health, substance use, personal safety, perceived stress, and social support. Standardized healthy start (HS) tools, adapted for the local context, are used to ensure comprehensive and consistent assessment. The intake process emphasizes voluntary participation, confidentiality, and participant safety. Assessments are offered in one of three formats: in person at partner sites or in the participant’s home, by telephone, or virtual. The average assessment time is 30 to 45 min across modalities. To support participant comfort and engagement, motivational interviewing techniques are employed throughout the intake process.

#### 2.2.2. Assessment Tools

All participants complete a general intake form that includes items on community context, health coverage, general and reproductive health, mental health, substance use, interpersonal safety, and psychosocial stressors. During intake procedures, participants complete health forms originally developed by the HRSA and culturally adapted to the local language. These forms collect information on demographics and maternal and child health. Women with previous pregnancies provide additional information on pregnancy history. A tailored section for fathers also assesses paternal involvement during pregnancy and parenting. Pregnant participants complete a prenatal assessment form that gathers data on prenatal care, home safety, and breastfeeding intentions and includes a delivery outcomes section. This tool is used to identify health risks, promote access to prenatal care, and offer smoking cessation and mental health referrals. Recent additions to the form document the perceived utility of doula services during childbirth. Participants with infants or children under 18 months complete a parent–child assessment form, which collects sociodemographic data, birth outcomes, caregiving practices, safety, insurance status, and access to pediatric care. This instrument is used to support safe sleep promotion, breastfeeding education, and developmental monitoring.

#### 2.2.3. Psychosocial and Behavioral Screening

Participants also complete a compilation of validated instruments to assess mental health, substance use, and parenting risk. These include the Generalized Anxiety Disorder 7-item scale (GAD-7); Patient Health Questionnaire-9 (PHQ-9); Depression Scale for Males; Alcohol, Smoking and Substance Involvement Screening Test (ASSIST); Adult Adolescent Parenting Inventory–2 (AAPI-2); Relationship Assessment Tool (RAT), formerly known as WEB; and the Adverse Childhood Experiences (ACE) Questionnaire. These tools inform service planning, referral decisions, and risk stratification, particularly for individuals with elevated psychosocial or behavioral health needs. All screening procedures adhere to the recommended clinical guidelines and scoring thresholds established for each validated tool.

### 2.3. Qualitative Data Collection and Analysis

Qualitative data were collected solely through open-ended questions included in the standardized intake assessments administered to participants during enrollment. These questions captured participants’ perceptions, feelings, and expectations regarding their health, care experience, and plans. Responses were recorded in writing by program staff during the intake process or during follow-up meetings. A simple thematic analysis was conducted on these written responses using an inductive approach to identify recurring patterns and themes. The final themes reflected key psychosocial domains such as empowerment, gratitude, anxiety, and health-related intentions.

## 3. Results: Program Outcomes and Impact

Between 2019 and 2023, the *Comienzo Saludable* Puerto Rico program showed changes across multiple maternal and child health indicators. Improvements were observed in rates of postpartum visit completion, depression screening and referral, reproductive life planning, and father/partner engagement. The program also tracked changes in preterm births, low birth weight, and infant mortality among enrolled participants during this period. The data presented in subsequent tables and figures summarize these trends and coverage rates across key domains of service delivery.

### 3.1. Sociodemographic Characteristics

A total of 1134 participants were enrolled in the program and classified into the following categories: preconception women (10, 0.9%), pregnant women (763, 67.3%), postpartum mothers (111, 9.8%), interconception women (83, 7.3%), and fathers or partners (167, 14.7%). These categories underscore the program’s commitment to a family-centered model of care, engaging not only birthing individuals but also male partners and other caregivers to strengthen social support structures—known determinants of positive maternal and newborn outcomes.

Most participants were women (85.3%) who entered the program during pregnancy or the postpartum period. Male participants, primarily fathers and partners (14.7%), were engaged through tailored, family-centered strategies.

The majority of participants were between 20 and 34 years old (81.4%), reflecting a typical childbearing age distribution. Adolescents aged 15–19 made up 5.8% of the sample, a group at elevated risk for adverse perinatal outcomes. Participants over 35 years of age accounted for 12.9% and were noted for higher clinical risks associated with advanced maternal age. These age-based distributions informed targeted support strategies, particularly for younger and older pregnant individuals. Additional risk groups identified in the population included non-pregnant caregivers and male partners, each with distinct psychosocial and health needs requiring tailored engagement.

Educational attainment varied: 20% had completed high school, and 38.3% had some college or a post-secondary degree. Regarding employment, 25.6% of participants were unemployed or engaged in informal/part-time work, and 7.1% were students, primarily among younger age groups (<25 years). These indicators of economic vulnerability reinforce the importance of integrating social determinants of health into maternal care approaches.

### 3.2. Impact on Maternal, Paternal, Newborn, and Child Health

Table 1 presents trends in three key infant health outcomes among Healthy Start Initiative (HSI) participants from 2019 to 2023 compared to the 2020 community baseline. The percentage of low-birth-weight infants among HSI participants declined from 6.25% in 2019 to 6.76% in 2023, representing a 9.59% decrease from 2019 and a 27.21% improvement relative to the community baseline (10.21%). Preterm births dropped significantly from 13.54% in 2019 to 6.80% in 2023, a 29.99% reduction and 30.94% lower than the community benchmark (11.34%). Most notably, the infant mortality rate among HSI participants declined from 6.25% in 2019 to 0% by 2021, with sustained zero mortality through 2023. This reflects a 12.5% reduction from 2019 and a dramatic 75% improvement relative to the community baseline (8.89%). These trends suggest measurable improvements in perinatal outcomes among program participants over the five-year period.

#### 3.2.1. Impact on Maternal Health

Table 2 presents longitudinal data on selected maternal and paternal health indicators collected through the *Comienzo Saludable* Puerto Rico initiative between baseline (pre-implementation in 2019) and 2024. Indicators include rates of depression screening and referrals, postpartum visit completion, reproductive life planning, tobacco cessation practices, and engagement of fathers/partners during pregnancy and early childhood. Additional indicators reflect services related to interconception care, intimate partner violence screening, access to usual sources of medical care, and well-woman visits. These data illustrate changes in service delivery performance and client engagement over time and support an assessment of the program’s impact on family-centered maternal health outcomes.

Annual monitoring revealed consistent increases in screening and referral rates for maternal depression, reproductive life planning, and paternal involvement. Indicators such as depression screening and referral rates, reproductive life planning, and father/partner involvement showed upward trends. Notably, the proportion of participants receiving depression referrals increased from 89.6% at baseline to 100% in 2024. Father and/or partner involvement with their children rose from 73.8% to 95.1% over the reporting period. Conversely, postpartum visit completion declined across the same timeframe, suggesting an area for targeted intervention.

#### 3.2.2. Impact on Newborn/Child Health

Improvements in infant health outcomes, including reduced infant mortality rates and better overall neonatal/child care, were achieved through Cuidado Compartido with healthcare providers. Child health indicators improved over the course of the program (Table 3). The proportion of infants ever breastfed increased from 84.1% at baseline to 95.5% in 2024. Safe sleep practices among caregivers rose from 44.4% to 61%. Rates of well-child visits and consistent medical care remained above 90% from 2021 onward.

#### 3.2.3. Health Gaps

From 2019 through early 2024, a total of 4096 validated mental health and developmental screeners were administered through the program (See Table 4). The Edinburgh scale (EPDS) was the most frequently utilized tool (n = 1696), reflecting its primary role in perinatal mental health surveillance. The GAD-7 was administered 1416 times, showing sustained application across years, with stable use between 2021 and 2023. The PHQ-9, applied primarily to non-pregnant and male caregivers, was used 343 times, suggesting more targeted deployment. Additionally, the Ages and Stages Questionnaire (ASQ), used to assess child development, showed a sharp increase in use beginning in 2020, peaking in 2022 (n = 210) and totaling 641 administrations overall.

Annual screening volume was highest in 2020 (n = 1042), likely reflecting increased programmatic reach and heightened mental health concerns amid the COVID-19 pandemic (See Table 5). Notably, a sharp decline was observed in 2024 (n = 133), attributable to a three-month service interruption related to delays in program funding. This disruption underscores the sensitivity of mental health service delivery to systemic and financial contingencies, with potential implications for continuity of care.

Among prenatal and postpartum participants screened using the EPDS, 19.5% were identified as exhibiting possible depression symptomatology at some point during their program participation. Notably, a higher prevalence was observed among non-pregnant caregivers (i.e., those caring for children aged 6 to 18 months), 30.3% of whom screened positive for possible depression via the PHQ-9. In comparison, 11.1% of male caregivers—including fathers or male figures within the caregiving environment—screened positive for possible depression. Across all participant categories, 11.9% screened positive for possible anxiety symptomatology using the GAD-7, suggesting a moderate but noteworthy presence of anxiety-related symptoms within the caregiving population.

### 3.3. Healthcare Provider Outcomes

Training activities addressed parenting education, maternal and infant health, intimate partner violence, and communication (Table 6). Over 85% of healthcare provider participants reported high satisfaction with training. Additionally, new modules were introduced in 2024 to strengthen the competencies of community health workers, doulas, and care coordinators in priority regions.

### 3.4. Qualitative Findings: Participants’ Feelings and Plans

A thematic analysis was conducted using participant responses from open-ended survey items in participant intake and enrollment assessment tools. Two major domains are shared in this manuscript: feelings and plans.

#### 3.4.1. Feelings

Participants consistently expressed feelings of empowerment, gratitude, and renewed confidence in their ability to navigate healthcare systems. Many attributed this to the educational support, personalized services, and respectful care delivered by program staff:


*“I feel stronger and better prepared to take care of myself and my baby.”*



*“I’m thankful to have someone checking on me—it makes a big difference.”*



*“I’m scared about how I’ll manage when I go back to work.”*


Despite improvements, some participants reported persistent anxiety related to financial stress, balancing work and caregiving, and the continuity of care after discharge:


*“Even though I feel better, I’m scared about how I’ll manage when I go back to work.”*


#### 3.4.2. Plans

Participants outlined clear intentions to sustain positive health behaviors, pursue educational or employment goals, improve family stability, and engage in community leadership:


*“I plan to keep all my baby’s appointments and mine too—not just when something is wrong.”*



*“I want to finish my degree and find a job that gives me time with my child.”*



*“I want to help other women like me—if I can do it, they can too.”*



*“We are saving to move into a safer neighborhood for our baby.”*


These narratives underscore the program’s success in promoting both individual agency and broader family well-being beyond clinical indicators.

## 4. Discussion

The observed improvements in maternal and child health indicators suggest that the *Comienzo Saludable* Puerto Rico program may have contributed meaningfully to reducing modifiable risks and addressing structural barriers affecting perinatal health outcomes. The program’s culturally tailored, community-based approach—anchored in the *Cuidado Compartido* (Shared Care) model—facilitated targeted efforts in areas such as depression screening and referral, postpartum care follow-up, reproductive life planning, and father/partner engagement. These strategies aligned with key domains of comprehensive maternal and infant well-being and were especially relevant in underserved communities with limited access to coordinated care.

Both quantitative and qualitative findings underscore the effectiveness of this integrated approach. Improvements across key indicators—including depression screening and referral rates, breastfeeding initiation, father/partner involvement, and well-childcare—highlight the program’s success in addressing both medical and psychosocial needs. These outcomes align with existing evidence demonstrating the positive impact of coordinated, comprehensive care on improving maternal and infant health, particularly among underserved populations [21,22,23].

The *Comienzo Saludable* Puerto Rico program demonstrates how an integrated, community-based Networks of Care model can significantly improve maternal and infant health outcomes. Key perinatal indicators—including low birth weight, premature birth, and infant mortality—showed measurable improvement from 2019 to 2023:Low birth weight among participants declined from 6.25% in 2019 to 6.76% in 2023, representing a 27.2% improvement compared to the community baseline (10.21%).Premature birth rates dropped from 13.54% to 6.80%, a 30.9% reduction compared to the baseline (11.34%).Infant mortality among children receiving Healthy Start services fell from 6.25% to 0%, representing a 75% reduction compared to the community baseline (8.89%).

These improvements reflect the combined impact of care coordination, culturally tailored interventions, and consistent community engagement. Although causality cannot be confirmed, the observed trends align with known pathways between enhanced prenatal care, maternal support systems, and improved birth outcomes. One of the program’s most notable achievements was the increase in depression screening and referrals, reaching 100% referral rates by 2024. This reflects the improved detection of perinatal mental health needs and the program’s capacity to provide timely care.

Variations in depression prevalence—particularly among non-pregnant caregivers and male/father participants—highlight often-overlooked needs during early childhood development [9,24]. Anxiety symptoms, though less frequent, further underscore the importance of comprehensive mental health support.

Mental health screening volume peaked in 2020, likely driven by pandemic-related stress and heightened awareness of mental health needs. The peak in depression screening volume observed in 2020 likely reflects both the program’s full-scale implementation—enabling more consistent screening across all eligible participants—and increased public and provider awareness of mental health concerns during the early COVID-19 pandemic. However, coverage declined from 66.7% in 2020 to 47.8% in 2023. This decline may reflect operational challenges such as staff turnover, fluctuating caseloads, post-pandemic service disruptions that could be attributed to funding delays, highlighting the vulnerability of these services to financial instability. This emphasizes the need for sustained and reliable funding to ensure continuity in screening and referral systems that support long-term caregiver and family well-being. Standardized assessments—such as the PHQ-9, GAD-7, ASQ, and others—enabled the timely identification of perinatal depression, anxiety symptoms, developmental delays, and substance use risks, helping tailor services for high-need participants.

Behavioral improvements were also observed in breastfeeding practices, safe sleep behaviors, and father engagement—areas linked to consistent home visits, parenting education, and culturally tailored support. High rates of partner involvement, especially in caregiving roles, further demonstrate the program’s effectiveness in engaging fathers and male caregivers. This is significant given the well-established associations between father engagement and positive maternal and child health outcomes [9,24]. The program’s inclusive approach—supporting all caregivers, not just birthing individuals—positions it as a model for comprehensive and equitable perinatal care.

Despite these successes, interconception care tracking began in 2020, which is why the 2019 baseline is recorded as 0%. The high screening percentage in 2019 reflects a small initial cohort, and subsequent lower values correspond to a broader participant base. These fluctuations highlight the importance of sustained workforce capacity and program continuity to ensure consistent service delivery over time.

Finally, declines in postpartum visit completion rates point to persistent barriers. Participants with full-time employment frequently cited limited maternity leave, inadequate childcare, transportation issues, and inflexible schedules. These findings suggest a need for adaptive and innovative follow-up strategies such as mobile health, home visits, and flexible clinic hours, especially for adolescents and older mothers who face distinct clinical and structural risks [25].

In sum, *Comienzo Saludable* Puerto Rico serves as a compelling case study for how locally driven, continuous, community-integrated care can address entrenched maternal and child health barriers. Ensuring the program’s long-term sustainability through consistent funding and systemic support will be essential to preserving these gains and expanding their reach.

### 4.1. Strengthening Networks of Care Through Emotional Support and Empowerment

The qualitative findings added critical depth to the interpretation of quantitative outcomes. Participants consistently reported increased confidence, agency, and a sense of empowerment in navigating health systems—key psychosocial outcomes often underrepresented in program evaluations. Feelings of gratitude, pride, and resilience were frequently linked to the program’s emphasis on building continuous and trusting relationships among families, healthcare providers, and community organizations. Many participants articulated aspirations extending beyond immediate health concerns, including educational and career goals, family stability, and community advocacy. These insights suggests that the program may serve as a catalyst not only for improved health outcomes but also for broader social transformation within families and communities.

These findings align with prior research showing that relational continuity—a core element of the Networks of Care model—enhances health outcomes by promoting patient engagement, adherence to care plans, and proactive health behaviors [26]. Notably, the program’s focus on culturally sensitive, personalized support contributed to a sense of belonging within the care system, helping to overcome common barriers to maternal and newborn health among marginalized populations.

The integration of extensive service delivery—over 15,000 individual services and nearly 500 group sessions—alongside a robust capacity-building framework further expanded the program’s reach. High levels of participant satisfaction with training, particularly among community health workers and doulas, suggest that investments in workforce development contributed to improved care delivery and participant trust.

### 4.2. Implications for Policy and Practice

The results of this evaluation provide compelling evidence that community-anchored care networks can effectively address maternal and newborn health outcomes, particularly in regions characterized by resource constraints and systemic inequities. Key program strategies—such as the integration of community health workers, prioritization of postpartum mental health, and enhancement of case navigation services—offer invaluable insights for the design and implementation of similar initiatives across U.S. territories and rural communities.

Furthermore, the qualitative findings highlight the importance of measuring emotional and aspirational outcomes alongside traditional health indicators. Such holistic evaluation practices are essential for capturing the full impact of health interventions on vulnerable populations.

The marked reductions in low birth weight, preterm birth, and infant mortality underscore the value of the Cuidado Compartido model as a replicable strategy for addressing persistent perinatal health inequities. These findings support the need for the broader adoption of integrated care approaches in similarly underserved areas across the U.S. and its territories. The model’s success in Puerto Rico offers a strong policy case for embedding Networks of Care in public health maternal–child programs nationwide.

### 4.3. Challenges and Barriers

Despite its critical contributions, *Comienzo Saludable* operates within a landscape of persistent structural and logistical challenges that impact both service delivery and long-term sustainability.

Access to quality maternal healthcare remains limited in many service areas due to hospital closures, workforce shortages, and inadequate infrastructure. Pregnant individuals are often required to travel long distances—sometimes several hours—to reach delivery facilities, increasing the risk of complications. A shortage of OB/GYNs committed to evidence-based care further restricts access, particularly in rural and underserved regions.

Institutional barriers also impede the integration of doulas, who are frequently denied access to labor and delivery rooms despite evidence supporting their positive impact on birth outcomes. This exclusion diminishes emotional support for birthing individuals and delays critical early postpartum practices, such as skin-to-skin contact and breastfeeding, often due to restrictive hospital policies.

Cultural and linguistic differences, along with deep-rooted mistrust in institutions, can further hinder healthcare engagement. While *Comienzo Saludable* employs culturally responsive staff and tailored strategies to build trust, addressing these barriers requires ongoing and intentional effort.

Program sustainability is another major challenge. Inconsistent funding threatens staffing, service continuity, and the program’s ability to scale in response to community needs. Infrastructure limitations—such as unreliable electricity and internet connectivity—particularly in remote areas, disrupt virtual care delivery, documentation, and communication. Power outages especially hinder access to telehealth, preventing consistent engagement between staff and families.

Delivering in-person services across geographically dispersed areas remains both costly and time-intensive. While virtual care presents a promising alternative, technological barriers and frequent service interruptions limit its effectiveness.

These challenges highlight the urgent need for broader structural reforms. Integrating community-based birth workers into formal healthcare systems, updating hospital policies to reflect evidence-based maternal health practices, and aligning care with global standards are essential for improving maternal health outcomes. Despite these hurdles, *Comienzo Saludable* continues to provide impactful, community-driven care. Sustained investment and aligned policy support are critical to ensuring this essential work reaches those who need it most.

While the program successfully addressed barriers such as care coordination, mental health screening, and postpartum follow-up, many challenges—particularly those related to provider shortages, transportation in rural areas, limited maternity leave, childcare access, and inflexible work schedules—remain beyond its direct scope. These systemic barriers highlight structural limitations within the broader health system that require policy reforms, infrastructure investment, and multisector collaboration at territorial and federal levels. Future programmatic efforts may benefit from strategic partnerships to advocate for solutions that promote maternal health and sustained postpartum engagement.

### 4.4. Limitations and Future Directions

As an implementation science evaluation of a real-world maternal and child health service program, this study is subject to several limitations inherent in practice-based research. First, the absence of a control or comparison group limits the ability to establish causal inferences between program participation and observed changes. However, as the primary objective was to evaluate outcomes within a service delivery context, the focus was on capturing real-world effectiveness and feasibility. Future evaluations may benefit from incorporating quasi-experimental or matched comparison designs to strengthen attribution while maintaining ecological validity.

Second, data completeness varied across indicators, particularly for postpartum visit rates, which may have affected trend accuracy and interpretability. These gaps often reflected structural barriers faced by participants, such as difficulties accessing services due to work, transportation, or caregiving responsibilities.

Third, while the qualitative findings were rich and informative, they were derived from self-reported responses to open-ended intake questions. While not intended to be statistically representative, these narratives offer meaningful insights into participants lived experiences and service interactions. Consistent with the interpretivist nature of qualitative inquiry, these data illuminate the subjective and relational dimensions of care. However, as with most self-reported data, responses may be influenced by recall bias or social desirability, which could shape how experiences are described.

Attrition posed another challenge. Given the program’s extended duration (ranging from 6 to 27 months), participant dropout may have introduced selection bias over time. Lastly, persistent logistical constraints, including transportation barriers and limited-service infrastructure in rural areas, continued to constrain service reach in some municipalities. Notwithstanding these limitations, the *Comienzo Saludable* model offers a replicable and culturally grounded framework for integrative maternal and child healthcare. By prioritizing family engagement, mental health integration, and community participation, the program addresses entrenched health challenges in Puerto Rican communities.

Looking ahead, future efforts may benefit from further evaluation of the program’s scalability, sustainability, and potential alignment with broader policy frameworks, particularly in similarly underserved regions. Future research should also examine longitudinal outcomes related to participants’ educational, career, and leadership trajectories to better understand the sustained impact of community-based Networks of Care interventions such as the *Cuidado Compartido* model.

## 5. Conclusions

The *Comienzo Saludable* Puerto Rico program exemplifies how a culturally adapted Networks of Care framework—“*Cuidado Compartido*”—centered on community integration, continuity of care, and community empowerment can significantly improve maternal and newborn health outcomes while fostering emotional well-being, resilience, and long-term empowerment. Its emphasis on reducing infant mortality, preterm births, and low birth weight aligns with national and global public health priorities, particularly in territories experiencing high rates of these indicators like Puerto Rico. This model offers a compelling, scalable strategy for addressing health gaps and differences and holds considerable promise for replication in other underserved populations, both within and beyond Puerto Rico.

By addressing both clinical and social determinants of health, the program not only reduced adverse birth outcomes but also fostered long-term emotional well-being, caregiver confidence, and sustained family engagement. These achievements are particularly noteworthy given the complex health inequities, healthcare access barriers, and resource limitations faced by many communities on the archipelago.

The *Comienzo Saludable* Puerto Rico program offers compelling evidence that community-anchored, culturally grounded Networks of Care can achieve substantial improvements in maternal and newborn health outcomes. From 2019 to 2023, the program achieved a 27.2% reduction in low birth weight, a 30.9% drop in preterm births, and a 75% decrease in infant mortality relative to community baselines. Its success underscores the value of investing in community-based systems of care that support families throughout the perinatal period. As such, it holds considerable promise for adaptation in other underserved and high-risk populations across U.S. territories and beyond.

These results highlight the promise of implementation science to inform real-world care models that are responsive to local contexts. The *Cuidado Compartido* framework serves as a valuable template for maternal–child health programs seeking to integrate clinical services, emotional support, and community participation.

## Figures and Tables

**Figure 1 ijerph-22-01204-f001:**
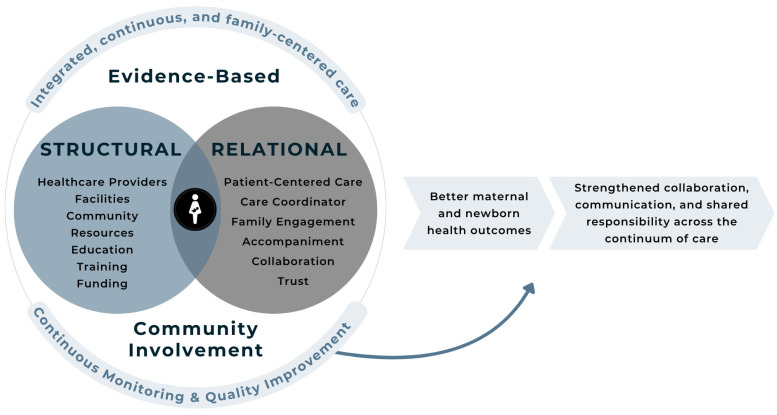
Image of the Cuidado Compartido Model—*Comienzo Saludable* Puerto Rico. Modified from Kalaris et al. [7].

**Figure 2 ijerph-22-01204-f002:**
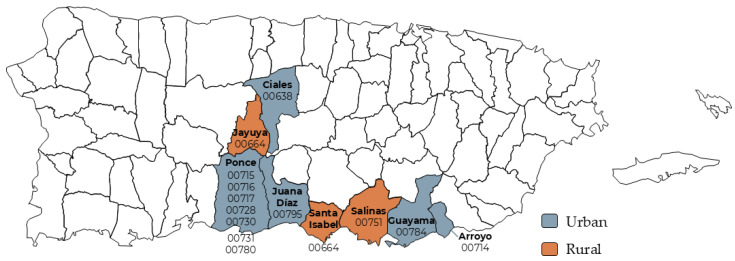
Geographic reach of *Comienzo Saludable* Puerto Rico program sites. Highlighted areas represent urban and rural municipalities in the central-southern region of Puerto Rico where maternal and child health services were delivered between 2019 and 2023.

**Figure 3 ijerph-22-01204-f003:**
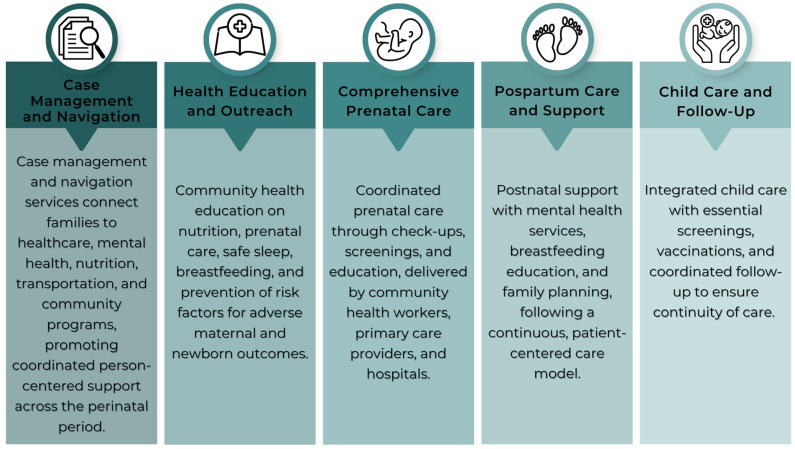
Main service components of *Comienzo Saludable*.

**Table 1 ijerph-22-01204-t001:** Key infant health outcomes among Healthy Start Initiative (HSI) participants from 2019 to 2023.

Indicator	Community Baseline ^1^ (2020)	Healthy Start Initiative (HSI) Participants							Percent Change 2023 Against Community Baseline ^4^
		2019 ^2^	2020	2021	2022	2023	Trend	Percentage Change in HSI in 2019 and 2023 ^3^	
Low Birth Weight among Children receiving Healthy Start Services before Birth	10.21%	6.25%	5.58%	7.12%	7.35%	6.76%	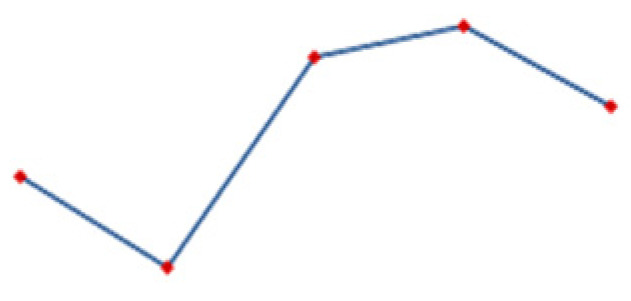	−9.59%	−27.21%
Premature Births among pregnant participants receiving Healthy Start Services before Birth	11.34%	13.54%	11.49%	8.30%	10.69%	6.80%	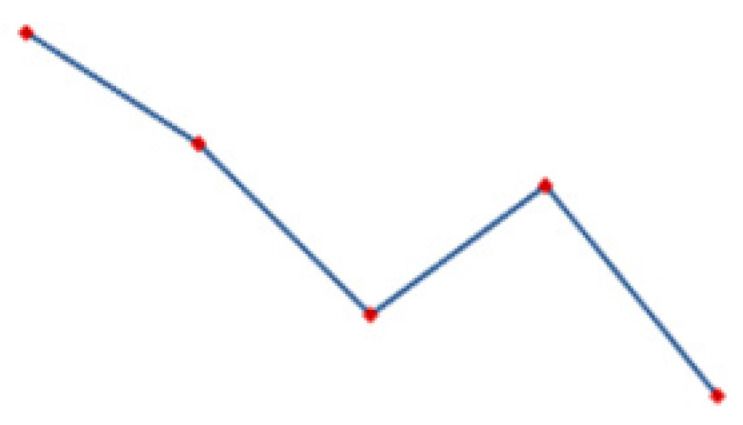	−29.99%	−30.94%
Infant Mortality Rate of Children receiving Healthy Start Services before Birth	8.89%	6.25%	0.66%	0.00%	0.00%	0.00%	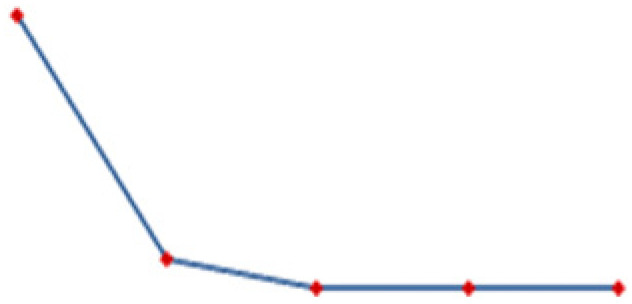	−12.50%	−75%

^1^ Data Source. ^2^ Healthy Start Initiative Participant Data. ^3^ (CY2023-CY2019)/CY2019. ^4^ (CY2023-Community Baseline)/Community Baseline.

**Table 2 ijerph-22-01204-t002:** Annual maternal and paternal health outcomes (2019 baseline and changes over time 2020–2024).

Indicators	Baseline *	2020	2021	2022	2023	2024
	132	421	524	579	549	263
Depression screening	123/123 (100%)	371/372 (99.7%)	445/454 (98%)	491/493 (99.6%)	470/472 (99.6%)	230/230 (100%)
Depression referral	26/29 (89.6%)	56/60 (93.3%)	68/70 (97.1%)	58/79 (73.4%)	41/77 (53.2%)	26/36 (72.2%)
Father and/or partner involvement during pregnancy	47/54 (87%)	180/204 (88.2%)	276/292 (94.5%)	264/288 (91.7%)	239/258 (92.6%)	56/63 (88.9%)
Father and/or partner involvement with their child	48/65 (73.8%)	214/237 (90.3%)	274/267 (93%)	332/352 (94.3%)	340/360 (94.4%)	174/183 (95.1%)
Interconception period (18 months between pregnancies)	0/0 (0%)	2/3 (66.7%)	29/116 (25%)	47/134 (35.1%)	40/116 (34.5%)	11/23 (47.8%)
Intimate partner violence screening	123/123 (100%)	376/380 (98.9%)	448/454 (98.7%)	494/494 (100%)	472/473 (99.8%)	230/231 (99.6%)
Postpartum visit (first month postpartum)	12/14 (85.7%)	33/83 (39.8%)	59/233 (25.3%)	72/322 (22.4%)	57/207 (27.5%)	-
Reproductive life plan	50/123 (40.6%)	292/312 (93.6%)	363/454 (80%)	257/461 (55.7%)	368/473 (77.8%)	221/231 (95.7%)
Tobacco use during prenatal stage	27/27 (100%)	47/47 (100%)	201/203 (99%)	184/187 (98.4%)	351/359 (97.8%)	134/135 (99.3%)
Usual source of medical care	98/109 (89.9%)	355/355 (100%)	274/452 (60.6%)	312/471 (66.2%)	351/458 (76.6%)	185/223 (83%)
Well-woman visit	95/114 (83.3%)	372/374 (99.5%)	418/457 (91.5%)	455/492 (92.5%)	443/470 (94.3%)	220/230 (95.7%)

* Descriptive trends compared to the 2019 community baseline. Values represent percentages and changes over time among program participants. No inferential statistical tests were applied; data are presented for descriptive and implementation-monitoring purposes only.

**Table 3 ijerph-22-01204-t003:** Annual progress on child health indicators (2019 baseline and changes over time 2020–2024).

Indicators %	Baseline	2020	2021	2022	2023	2024
	80	287	306	368	368	186
Breastfed ever (0–12 months)	53/63 (84.1%)	182/187 (97.3%)	192/207 (92.8%)	278/289 (96.2%)	296/312 (94.9%)	147/154 (95.5%)
Breastfeeding at 6 months (6–12 months)	6/20 (30%)	19/29 (65.5%)	45/71 (63.4%)	92/129 (71.3%)	45/85 (52.9%)	21/41 (51.2%)
Safe sleep practices (0–12 months)	8/18 (44.4%)	107/187 (57.2%)	36/77 (46.8)	159/331 (51.1%)	172/304 (56.6%)	97/159 (61%)
Regular source of medical care	63/74 (85.1%)	236/254 (92.9%)	233/275 (84.7)	335/354 (94.6%)	354/366 (96.7%)	179/186 (96.2%)
Well-child visit	54/70 (77.1%)	219/238 (92%)	189/210 (90%)	282/311 (90.7%)	309/341 (90.6%)	158/174 (90.8%)

Descriptive trends compared to the 2019 community baseline. Values represent percentages and changes over time among program participants. No inferential statistical tests were applied; data are presented for descriptive and implementation-monitoring purposes only.

**Table 4 ijerph-22-01204-t004:** Screenings administered by type and by year.

Validated Screening Tool	2019	2020	2021	2022	2023	2024 *	Total
ASQ	0	5	152	193	210	81	641
Edinburgh	120	575	344	338	296	23	1696
GAD	131	351	296	312	310	16	1416
PHQ-9	41	111	68	39	71	13	343
Depression Scale for Males	0	0	0	13	9	0	22
Total measures administered	292	1042	860	895	896	133	4118

* In this year a three-month gap was experienced due to delays in award/funding.

**Table 5 ijerph-22-01204-t005:** Amount of individual and group services provided by year.

Service	2019	2020	2021	2022	2023	2024 *	Total
Breastfeeding support and education	22	317	21	37	71	13	481
Doulas	2	202	815	1170	807	208	3204
Father individual encounter	2	51	15	106	4	0	178
Parenting Fundamentals	5	124	84	100	70	0	383
GGK Growing Great Kids	0	111	247	192	185	45	780
Home visit	52	723	1837	1963	1565	509	6649
Individual service plan	74	272	235	253	221	4	1059
Mental health session	36	298	212	217	363	72	1198
Father figure group session	0	32	12	9	15	0	68
Parenting Fundamentals group session	7	91	55	69	56	0	278
Prenatal group session	0	8	42	37	30	2	119
Other group services	0	5	4	4	25	1	39

* In this year a three-month gap was experienced due to delays in award/funding.

**Table 6 ijerph-22-01204-t006:** Training and capacity-building activities (2020–2024).

Theme or Competencies	Sample Activities	Reach and Impact
CHW core training	Conducted standardized core competencies training for community health workers (CHWs), including modules on maternal mental health, reproductive life planning, safe sleep, and breastfeeding support	32 CHWs trained across seven municipalities; 100% completed core competencies; 87% increased confidence in maternal health knowledge
Parenting education	Growing Great Kids Curriculum©, Parenting Fundamentals Curriculum©, FatherIn15©	Delivered in cycles with up to 48 contact hours
Maternal and infant health practices	Breastfeeding certification, safe sleep, perinatal mental health, birth preparation	78 contact hours; >85% satisfaction
Violence and crisis response	Domestic and sexual violence, crisis intervention, IPV legal aspects	Response rates > 85%; high satisfaction reported
Communication and education	Effective communication, engaging fathers, team integration	Broad CHWs, doulas, and care coordinators
Leadership development	Coaching, supervision training, HR policy workshops	Two regional cohorts in 2024
Technology and tools	Canva training, data system revisions, Fathering 15©, ASQ and AAPI form use	Enhanced monitoring and reporting capacity

## Data Availability

Data supporting the findings of this study are available upon reasonable request from the corresponding author, subject to program confidentiality guidelines and institutional data-sharing policies.

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
