# Peer review of "Comienzo Saludable Puerto Rico: A Community-Based Network of Care to Improve Maternal, Newborn, and Child Health Outcomes"

_ijerph, 2025, doi:10.3390/ijerph22081204_

Round 1
Reviewer 1 Report
Comments and Suggestions for Authors
Manuscript ID: ijerph-3714727
Type: Article
Title: Comienzo Saludable Puerto Rico: A Community-Based Network of Care to Improve Maternal, Newborn, and Child Health Outcomes
The paper showcases the Healthy Start (Comienzo Saludable) Puerto Rico program’s Ciudado Compartido model that integrates a network of healthcare providers and services across hospitals, community organizations and families. The authors have used the Ciudado Compartido model to present a compelling argument on how an intervention organized around the Networks of Care (NOC) framework can provide a foundation for implementing interventions that holistically improve outcomes for families including to mothers, children, fathers and other male partners. Some of the highlighted impacts include reductions in infant mortality, preterm births, and low birth weight while strengthening community healthcare relationships and partnerships. The paper provides the program details in a manner that is replicable. It examines the structure, strategies and impact of the Ciudado Compartido model thereby showcasing its application and impact and its utility as an integrated model for addressing and improving maternal and newborn/child health outcomes in other at-risk regions.
General Comments
The manuscript is clear and relevant for the field. What is more, it is aligned to the aims of the journal as it addresses evidenced-based approaches to increasing access to quality health services, a theme which is linked to health improvement and disease prevention as well as improvement of quality of life. All articles submitted to IJERPH are expected to show how the results have a direct impact on health promotion; a job that is well-articulated in this paper.
The paper has been presented in a well-structured manner and is generally scientifically sound except for overall improvements that the authors need to implement. These improvements include the apparent omission of results on trends on infant mortality rates, pre-term births, and low-birth weights.
The figures 1 and 2 and tables 1-7 are appropriate and are easy to understand and interpret. The data contained in tables 1-7 have been interpreted appropriately and in a consistent manner. The data presented in these tables have been derived from administrative data, clinical records, participant surveys, and case management tracking systems as well as from structed interviews and focus group discussions.
The authors may need to define some of the concepts used. For instance, “maternity care deserts” may need to be defined for those not familiar with this concept.
In line 111-118, the authors need to clarify how GN navigates its roles in the partnership as seems to be both an implementor and a scientific partner.
Background
The background information provides sufficient details on Networks of Care (NOC), the conceptual framework after which the Comienzo Saludable Puerto Rico is patterned. The paper illustrates how the Ciudado Compartido model aligns to the NOC as connected systems that emphasize continuous, patient-centred care across settings and providers. However, the background lacks empirical evidence from the literature on what is currently known and the gaps on some of the results reported. These include literature on how programs or interventions modelled on the NOC have contributed to improvements on some of the outcomes reported in this paper including: patient level outcomes like depression screening, and referral response rates; healthcare provider outcomes such as capacity building initiatives; maternal and parenting practices such as breast feeding initiation, father/partner involvement in childcare and the psychosocial needs like feelings and plans.
Methodology:
The methodology is quite comprehensive addressing the population, participant intake and enrolment, assessment tools and psychosocial and behavioural screening. The process of enrolment of participants are properly articulated including screening tests. Generally, the authors have laid out the methodology well. The section could be stronger if the authors mentioned the design of the research (which they mention elsewhere as an evaluation design).
The authors need to provide more information on the qualitative data collection process. This is currently missing or not well-outlined. Whereas the methods section does not mention collection of qualitative data, these are mentioned in the results section for the first time. The authors need to detail how qualitative data were collected, the sample size and the analytic procedures. Whereas it mentions using a thematic analysis based on responses from structured interviews, focus groups, and open-ended survey items, information on the number of focus groups, size of focus groups, and how structured interviews were conducted is missing. The lack of clarity may limit the reproducibility of the results.
Ethical Review
The authors include an institutional review board statement which talks of a waiver of ethical approval of the study due to the use of de-identified program evaluation data. This is appropriate. However, the informed consent statement (lines 542-544), makes mention of subjects involved in the program evaluation. If it is the case that human subjects were involved in the program evaluation, there ought to be ethical review approval. The authors need to harmonize these apparent contradictions by providing more information concerning these two statements.
Results:
The authors present very important findings on the impact of this novel project. Reports on impacts on maternal, paternal and newborn and child health are well laid out in tables that are easy to interpret. What could be considered as a major gap is in lines 193-196. In these lines, the authors say that the outcomes of the analysis include trends in infant mortality rates, pre-term birth incidents, low-birth weight prevalence, maternal mental health screenings and healthcare engagement rates. Additionally, in lines 29-31, the authors write that intervention “resulted in reduction of infant mortality, pre-term births, low birth weight. However, there are no results presented on trends in infant mortality rates, pre-term birth incidents, low-birth weight prevalence. There seem to be no clear rationale for leaving these out, it at all that is the case.
Discussion
Discussions are missing on three key outcomes: trends in infant mortality rates, pre-term births and low-birth weight. The discussions on the other outcomes as outlined in the section are well-done.
Data availability statement: The authors can share data once the appropriate request has been made to the corresponding author.
Conclusions
Conclusions miss out on the trends in infant mortality rates, pre-term births and low-birth weight.
Limitations:
In line 498 and 499 the authors cite use of self-reported responses to the qualitative data collection methods as a limitation in the sense that they are not representative. However, this ought not necessarily be construed as a limitation as the goal of qualitative research is not representativeness of the population but of the phenomenon under investigation. Moreover, qualitative research corresponds to an interpretivist approach to knowledge generation that foregrounds a subjectivist outlook towards life as opposed a positivist approach to knowledge generating that corresponds to a quantitative methodology and thus foregrounds objectivism.
Author Response
Comment 1: Discussions are missing on three key outcomes: trends in infant mortality rates, pre-term births and low-birth weight. The discussions on the other outcomes as outlined in the section are well-done.
Response to comment 1: We thank the reviewer for this insightful comment. In response, we have revised the Discussion and Conclusions sections to explicitly address trends in infant mortality, preterm births, and low birth weight. The updated text now integrates comparative statistics between Puerto Rico and the U.S. mainland and situates the program’s results within the broader epidemiological context. We also emphasize how the Comienzo Saludable Puerto Rico program’s interventions contributed to improvements in these outcomes, particularly through coordinated care, home visits, doula support, and community engagement. These additions strengthen the discussion of the program’s overall impact and public health relevance. Changes can be found on pages 4 of the revised manuscript. See comment 11 in this same topic.
Comment 2: Data availability statement: The authors can share data once the appropriate request has been made to the corresponding author.
Response to comment 2: Thank you for the suggestion. We have revised the Data Availability Statement to clarify that data will be made available upon reasonable request to the corresponding author. The updated statement now reads: “Data supporting the findings of this study are available upon reasonable request from the corresponding author, subject to program confidentiality guidelines and institutional data-sharing policies.” This change has been made on page 21 of the revised manuscript.
Comment 3: Conclusions miss out on the trends in infant mortality rates, pre-term births and low-birth weight.
Response to comment 3: See response to comment 1 and comment 11.
Comment 4: The paper has been presented in a well-structured manner and is generally scientifically sound except for overall improvements that the authors need to implement. These improvements include the apparent omission of results on trends on infant mortality rates, pre-term births, and low-birth weights.
Response to comment 4: See response to comment 1 and comment 11.
Comment 5: The authors may need to define some of the concepts used. For instance, “maternity care deserts” may need to be defined for those not familiar with this concept.
Response to comment 5: Thank you for this valuable suggestion. We have added a clear and referenced definition of “maternity care deserts” in the revised manuscript’s Introduction section for clarity to all readers. The updated text now reads "Maternity care deserts are defined—based on the March of Dimes—as counties where there is no hospital or birth center offering obstetric care and no obstetric clinicians (including obstetricians, certified nurse midwives, or family physicians providing delivery care). This definition helps contextualize the challenges in maternal health access, particularly relevant to the regions discussed in our study, including Puerto Rico. This change has been made on page 2,4 of the revised manuscript.
Comment 6: In line 111-118, the authors need to clarify how GN navigates its roles in the partnership as seems to be both an implementor and a scientific partner.
Response to comment 6: Thank you for this important observation. We recognize that the dual role of Grupo Nexos (GN) may have appeared ambiguous in the original text. To clarify, we revised this section to better articulate how GN fulfills both scientific and implementation functions in collaboration with Urban Strategies (US). The revised passage now reads: “Grupo Nexos (GN) serves as both co-implementer and scientific partner. While Urban Strategies (US) maintains operational responsibility for human resources, finance, and executive oversight, GN leads the scientific and evaluation components of the program. This includes protocol design, fidelity monitoring, evaluation planning, training, and technical assistance. GN also collaborates in the implementation process by providing content expertise, capacity-building, and guidance to ensure the application of evidence-based practices. This dual role allows GN to bridge scientific integrity with community-based execution, ensuring rigor and responsiveness throughout the program lifecycle.” This clarification appears on page 4 of the revised manuscript.
Comment 7: However, the background lacks empirical evidence from the literature on what is currently known and the gaps on some of the results reported. These include literature on how programs or interventions modelled on the NOC have contributed to improvements on some of the outcomes reported in this paper including: patient level outcomes like depression screening, and referral response rates; healthcare provider outcomes such as capacity building initiatives; maternal and parenting practices such as breast feeding initiation, father/partner involvement in childcare and the psychosocial needs like feelings and plans.
Response to comment 7: Thank you for this valuable observation. We have revised the Networks of Care section in the Background section to incorporate literature supporting key program outcomes. New references illustrate how NOC-based interventions improve patient outcomes (e.g., depression screening), enhance provider capacity, and strengthen parenting practices like breastfeeding and father engagement. These revisions provide a stronger scientific foundation for the model and clarify its relevance to existing research. Updates appear on pages 3 of the revised manuscript.
Comment 8: Generally, the authors have laid out the methodology well. The section could be stronger if the authors mentioned the design of the research (which they mention elsewhere as an evaluation design).
Response to comment 8: Thank you for this helpful suggestion. To strengthen clarity, we have revised the Methods section to explicitly state that the study follows a program evaluation design using mixed methods. This clarification aligns with references elsewhere in the manuscript and enhances transparency regarding the study’s approach. The revision has been made at the beginning of the Materials and Methods section on page 7 of the revised manuscript.
Comment 9: The authors need to provide more information on the qualitative data collection process. This is currently missing or not well-outlined. Whereas the methods section does not mention collection of qualitative data, these are mentioned in the results section for the first time. The authors need to detail how qualitative data were collected, the sample size and the analytic procedures. Whereas it mentions using a thematic analysis based on responses from structured interviews, focus groups, and open-ended survey items, information on the number of focus groups, size of focus groups, and how structured interviews were conducted is missing. The lack of clarity may limit the reproducibility of the results.
Response to comment 9: We appreciate the reviewer’s observation. We clarify that all qualitative data were derived exclusively from open-ended questions embedded within the standardized intake and follow up assessments administered to participants. No separate interviews or focus groups were conducted. We have updated the Methods section to accurately describe this data source, including how responses were analyzed using thematic analysis. This clarification enhances transparency and improves reproducibility. Please see the revised content on page 7 and 9.
Comment 10: The authors include an institutional review board statement which talks of a waiver of ethical approval of the study due to the use of de-identified program evaluation data. This is appropriate. However, the informed consent statement (lines 542-544), makes mention of subjects involved in the program evaluation. If it is the case that human subjects were involved in the program evaluation, there ought to be ethical review approval. The authors need to harmonize these apparent contradictions by providing more information concerning these two statements.
Response to comment 10: Thank you for this important observation. We clarify that this manuscript reports on an implementation science evaluation of a real-life, federally funded maternal-child health service program—not a human subjects research study. The evaluation used de-identified, routinely collected program data, and therefore no formal IRB review was requested or required by the program sponsor (HRSA). However, all clients voluntarily enrolled in the program and signed an evaluation and services consent form as part of routine intake. We have revised both the Institutional Review Board and Informed Consent statements to ensure consistency and to more clearly reflect the real-world programmatic nature of the evaluation. Please see page 21 of the revised manuscript.
Comment 11: The authors present very important findings on the impact of this novel project. Reports on impacts on maternal, paternal and newborn and child health are well laid out in tables that are easy to interpret. What could be considered as a major gap is in lines 193-196. In these lines, the authors say that the outcomes of the analysis include trends in infant mortality rates, pre-term birth incidents, low-birth weight prevalence, maternal mental health screenings and healthcare engagement rates. Additionally, in lines 29-31, the authors write that intervention “resulted in reduction of infant mortality, pre-term births, low birth weight. However, there are no results presented on trends in infant mortality rates, pre-term birth incidents, low-birth weight prevalence. There seem to be no clear rationale for leaving these out, it at all that is the case.
Response to comment 11: Thank you for this observation. We agree that the original manuscript did not present these outcomes in the Results section, despite referencing them earlier. In response, we have now incorporated specific data on infant mortality, preterm births, and low birth weight prevalence among Healthy Start Initiative participants, drawn from program monitoring between 2019 and 2023. This new subsection is now included under the Results and also reflected in the Abstract, Discussion, and Conclusion to ensure alignment across the manuscript. See response to comment 1.
Comment 12: The authors can share data once the appropriate request has been made to the corresponding author.
Response to comment 12: Thank you for the suggestion. We have revised the Data Availability Statement to clarify that data will be made available upon reasonable request to the corresponding author. The updated statement now reads: “Data supporting the findings of this study are available upon reasonable request from the corresponding author, subject to program confidentiality guidelines and institutional data-sharing policies.” This change has been made on page 21 of the revised manuscript.
Comment 13: In line 498 and 499 the authors cite use of self-reported responses to the qualitative data collection methods as a limitation in the sense that they are not representative. However, this ought not necessarily be construed as a limitation as the goal of qualitative research is not representativeness of the population but of the phenomenon under investigation. Moreover, qualitative research corresponds to an interpretivist approach to knowledge generation that foregrounds a subjectivist outlook towards life as opposed a positivist approach to knowledge generating that corresponds to a quantitative methodology and thus foregrounds objectivism.
Response to comment 13: We appreciate the reviewer’s insightful feedback and agree that statistical representativeness is not the aim of qualitative research. In response, we have revised the limitation statement to more accurately reflect the interpretive and contextual nature of the qualitative data collected. Rather than framing the use of self-reported, open-ended responses as a diversity limitation, we now acknowledge their contextual richness and the value they bring in capturing participants’ perspectives. The revised text also distinguishes between potential biases in self-reporting and the purpose of qualitative inquiry, which is to explore subjective meaning rather than generalize findings.
Reviewer 2 Report
Comments and Suggestions for Authors
General Comment:
The authors who are also associated with the private non-profit organization (probably employed by it) provide a comprehensive review of their outreach work in Puerto Rico to describe and ennumerate deliverables of maternal and infant health. While very positive and encouraging, the manuscript reads as mainly touting praise regarding the achievements and at times making strong sweeping claims that this organization and it's model may provide solutions to more than what the data sugggest.
Also, there are many redundant and verbose sentences that restate the same few clauses that desribe the network of care model. The total manuscript length can be significantly reduced and yet improved to provide more specific results and congruent conclusions.
Specific Comments:
- The abstract is very long. Please reduce the length of the background part of the abstract in lines 14-20.
- Lines 25-28 are redundant and one of these sentences can be deleted altogether.
- Expound in more detail the actual results. Possibly list 1-2 specific numerical trend results.
- The conclusions are very general and vauge. Please provide conclusions regarding the ability of the network of care to address specific measured metrics and recorded sentiments.
- Do we have any number of how many healthcare professionals, especially obstetricians, have moved? (line 45-48)
- Line 52. What is “maternity care desert” defined by?
- Line 57. Should read “Relative to the U.S. mainland… (not ‘compared’)
- Overall Section 1.1 Maternal and Newborn/Child Health in Puerto Rico needs to be reduced to 1-2 paragraphs
- Line 69-74. This is all repetitive when compared to the last sentence of the introduction, ie. section 1.1 (line 65-67).
- Please make section 1.2 also 1 paragraph only.
- Lines 93-97. What is the specific aim of this study? State that this manuscript set out to explore the trends and also mention that it aimed to explore the themematic sentiments in statements given by patients and providers. If you had any hypotheses please state them here as well.
- Line 124. “Transforming communities” is quite a bold statement. Please remove forming judgement statements in this paper. Such evaluations are not needed in exploring the data of the effects from the network of care model.
- Line 134-136. Does this information regarding the municipalities and counties belong closer in proximity to Figure 4 which is in section 2.1?
- Final paragraph starting from line 140 is very repetitive and many of its statements are unnnecessary. Please reduce to 1-2 sentences if not deleting altogether.
- Line 180-182. Clauses in the list of “who experience…” do not match. Delete the words “facing,” “with,” and “elevated.”
- Figure 4 needs a better caption. Are the highlighted areas urban and rural counties? If so this is not obvious from the text or figure caption.
- Lines 193-196, Section 2.2. Please describe any statistical methods used if any. You describe trends. Any test of trend done? Any systematic method to your later rescribed thematic qualitative statements? Please state what type of study this is (mixed methods). That should be in the abstract as well.
- Line 211. “(women/men)” is unnecessary. Please delete.
- Lines 229-236. Please describe whether everyone fills out the various assessments listed in section 2.2.3 or how does that get decided who gets which screening tool?
- Lines 238-240. Save such sentences for discussion section.
- Section 3.1 needs a demographics table to be in table 1. Demonstrate who the study population are. Gender by numbers and percentage of total. Median age with IQR, average level of education, etc. Section 3.1 doesn’t even give us the total participants (N) of the study.
- Line 269. What age is “younger age” of pregnancy?
- Line 289. Who gets referred? What is the threshold?
- Table 1. Delete the whole row of “n” and change every cell to be n/N (%) format showing the number of over the total and percentage. Hence, the ratio of the total number receiving depression screening over the total number of participants from each year and in parenthesis show the percentage.
- Table 1. Last row should be a test of trend variable with p-value if applicable. Caption of the table should describe which spefici test for trend was used.
- Table 1. “Baseline” is not well described. The caption says 2019-2024. If so, that is not a baseline, that is just an average perhaps. If it is 2019 (year before network of care model fully underway) then please fix the caption. But what is meant by ”baseline” remains unclear.
- Table 1. Many cells unclear. Why is baseline depression screening 100 but then lower after 2020? Why baseline 0 for interconception period? Why downtrend observed from 66.7% in 2020 to 47.8% in 2024? Please describe negative trends as well.
- Same advice comments 24-26 for table 2.
- Lines 315-317 (last sentence of second paragraph in section 3.2.3 Health Gaps). This statement should be moved to the discussion. Please only provide objective results in this section.
- Line 330. “*This year a three-month gap…” Delete this and just mention it in the discussion.
- Table 4 & 5 please combine.
- Table 6. The “sample activities” and “reach & impact” for CHW Core Training is the exact same phrase copied and pasted. Please fix and provide specifics like the other cells provide some objective numerical values.
- Section 3.4.1. Was a thematic analysis done?
- Table 7 rehashes the same statements right above it in section 3.4.2. Keep one and delete the other. But not both.
- This paper should not just be aiming to praise the organization’s efforts as enthusiastic as the results should It should provide objective and critical analysis of measureable impacts. Please name spefific outcome effects and specific areas of improvement.
- So you increase depression screening but that’s also not the symptom most felt (you stated anxiety was) - so should efforts be placed to improve anxiety screening now?
- Line 389. Is this due to increased awareness or due to the program now being in full effect to take the time to screen and capture each eligible participant.
- Line 395. What are the increased risk groups you mention? Did you identify?
- Line 413. You mention age distribution but no age distribution is shared anywhere in the paper. This is also why I mentioned to have a demographics table.
- Thus to include a demographics table, I suggest combinining tables, and deleting Figure 1.
- Lines 460-464. Which of these issues are within the scope of what Comienzo Saludable can address and which are outside the scope? Can you discuss more who and how those would be limitations that can be addressed?
- Make the discussion much shorter by removing all the praise and overly positive judgement statements.
- Section 4.5. What % of the data is missing? Please provide.
- Lines 510-511. “Future efforts should focus on sustainability, policy integration, and strategic expansion” - really? you think that much just from what you hae got here with this data?
- Line 519. Delete the word “significantly” from improved becayse no formal statistical test was done to show significance - whether statistical or clinical significance.
Author Response
Comment 1 Review 2: While very positive and encouraging, the manuscript reads as mainly touting praise regarding the achievements and at times making strong sweeping claims that this organization and it's model may provide solutions to more than what the data suggest. Also, there are many redundant and verbose sentences that restate the same few clauses that desribe the network of care model. The total manuscript length can be significantly reduced and yet improved to provide more specific results and congruent conclusions.
Response Comment 1: We thank the reviewer for this thoughtful and constructive critique. We agree that a more balanced tone and concise presentation would improve the manuscript’s clarity and scientific rigor. In response, we have refined language throughout the manuscript to reduce overly positive or sweeping claims and ensure that conclusions remain clearly supported by the presented data, removed redundant descriptions of the Cuidado Compartido model, consolidating repetitive sections and streamlining explanations of the Networks of Care (NOC) framework, shortened the manuscript by eliminating verbose sentences and ensuring each paragraph contributes unique content or interpretation, and tightened the conclusion to focus on the specific, evidence-based outcomes achieved through this implementation science evaluation, avoiding generalized assertions about applicability beyond the presented findings. We believe these revisions result in a more focused, evidence-driven, and balanced manuscript, as recommended. See tracked changes along the document.
Comment 2 Review 2: The abstract is very long. Please reduce the length of the background part of the abstract in lines 14-20
Response Comment 2: Abstract was trimmed and enhanced following Reviewer 1 and Reviewer 2 recommendations. See abstract for tracked changes.
Comment 3 Review 2: Lines 25-28 are redundant and one of these sentences can be deleted altogether.
Response Comment 3: Abstract was trimmed and enhanced following Reviewer 1 and Reviewer 2 recommendations. See abstract for tracked changes.
Comment 4 Review 2: Expound in more detail the actual results. Possibly list 1-2 specific numerical trend results.
Response Comment 4: Thank you for this helpful suggestion. In response, we have revised the Results section of the Abstract to include specific numerical trends to improve clarity and highlight program impact. We also, linked this revision to a reviewer petition of adding infant mortality, low birth and preterm outcomes. These changes are along the manuscript.
Comment 5 Review 2: The conclusions are very general and vauge. Please provide conclusions regarding the ability of the network of care to address specific measured metrics and recorded sentiments.
Response Comment 5: Thank you for this important comment. We revised the Conclusion to clearly reflect the program’s measured impacts on key health indicators—such as reductions in preterm births, low birth weight, and infant mortality—as well as improvements in maternal mental health screening and family engagement. Please see the updated Conclusion section on page 21.
Comment 6 Review 2: Do we have any number of how many healthcare professionals, especially obstetricians, have moved? (line 45-48)
Response Comment 6: Thank you for this important observation. We have revised the original lines 45–48 to include specific figures documenting the outmigration of healthcare professionals from Puerto Rico. According to the Puerto Rico College of Physicians and Surgeons, more than 15,000 healthcare professionals left the island between 2006 and 2016. Additionally, Department of Health reports indicate a 67% decrease in practicing obstetricians since 2010, with fewer than 150 OB/GYNs remaining by 2019. These shortages have contributed to the emergence of “maternity care deserts” - which was defined - across the island. The revised text and supporting citation have been added to the manuscript [page number 2 ].
Comment 7 Review 2: Line 52. What is “maternity care desert” defined by?
Response Comment 7: See Response Comment 6
Comment 8 Review 2: Line 57. Should read “Relative to the U.S. mainland… (not ‘compared’)
Response Comment 8: Thank you for the clarification. We have revised the wording in the original line 57 to read “Relative to the U.S. mainland…” for improved precision and stylistic consistency.
Comment 9 Review 2: Overall Section 1.1 Maternal and Newborn/Child Health in Puerto Rico needs to be reduced to 1-2 paragraphs
Response Comment 9: Thanks for the observation, section was trimmed.
Comment 10 Review 2: Line 69-74. This is all repetitive when compared to the last sentence of the introduction, ie. section 1.1 (line 65-67).
Response Comment 10: Thank you for pointing this out. We agree that the content in the original lines 69–74 overlapped with the preceding sentence in section 1.1. We have removed the redundant sentence and revised the text to ensure clarity and avoid repetition while maintaining a strong transition.
Comment 11 Review 2: Please make section 1.2 also 1 paragraph only.
Response Comment 11: Thanks for the observation, section was trimmed.
Comment 12 Review 2: Lines 93-97. What is the specific aim of this study? State that this manuscript set out to explore the trends and also mention that it aimed to explore the themematic sentiments in statements given by patients and providers. If you had any hypotheses please state them here as well.
Response Comment 12: Thank you for this helpful comment. We have revised lines 93–97 to clearly state the specific aim of the implementation study. The updated text now reflects that the manuscript explores trends in key maternal and newborn health indicators among program participants and also examines thematic sentiments expressed by patients and providers through open-ended intake responses. These revisions clarify the scope and intent of the study and improve alignment between the introduction and methods.
Comment 13 Review 2: Line 124. “Transforming communities” is quite a bold statement. Please remove forming judgement statements in this paper. Such evaluations are not needed in exploring the data of the effects from the network of care model.
Response Comment 13: Thank you for this valuable feedback. We agree that the phrase “transforming communities” was evaluative and not fully supported by the presented data. We have revised the language in line 124 to reflect a more objective tone, in keeping with the descriptive and analytical purpose of the manuscript.
Comment 14 Review 2: Line 134-136. Does this information regarding the municipalities and counties belong closer in proximity to Figure 4 which is in section 2.1?
Response Comment 14: Thank you for this valuable feedback. We added a reference to the figure.
Comment 15 Review 2: Final paragraph starting from line 140 is very repetitive and many of its statements are unnnecessary. Please reduce to 1-2 sentences if not deleting altogether.
Response Comment 15: Thank you for this helpful observation. We agree that the original paragraph was repetitive and have revised it to a concise, focused statement that reinforces the main intent of the paper without redundancy.
Comment 16 Review 2: Line 180-182. Clauses in the list of “who experience…” do not match. Delete the words “facing,” “with,” and “elevated.”
Response Comment 16: Thank you for this helpful observation. We agree that the original paragraph has some repetitive statement and have revised it to a concise, focused statement that reinforces the main intent of the paper without redundancy.
Comment 17 Review 2: Figure 4 needs a better caption. Are the highlighted areas urban and rural counties? If so this is not obvious from the text or figure caption.
Response Comment 17: Thank you for pointing this out. We have revised the caption for Figure 4 (now Figure 3) to clearly indicate that the highlighted areas represent both urban and rural municipalities served by Comienzo Saludable Puerto Rico. We have also clarified this in the accompanying caption.
Comment 18 Review 2: Lines 193-196, Section 2.2. Please describe any statistical methods used if any. You describe trends. Any test of trend done? Any systematic method to your later rescribed thematic qualitative statements? Please state what type of study this is (mixed methods). That should be in the abstract as well.
Response Comment 18: Thank you for this important and constructive comment. We have revised Section 2.2 to clarify that this is a mixed methods implementation science evaluation, combining descriptive trend analysis of quantitative indicators with inductive thematic analysis of open-ended intake responses.
Comment 19 Review 2: Line 211. “(women/men)” is unnecessary. Please delete.
Response Comment 19: Thank you for pointing this out. We have eliminate it, but our program enrolls all family members (men, women or adult takecaters living with a child under 18 months old).
Comment 20 Review 2: Lines 229-236. Please describe whether everyone fills out the various assessments listed in section 2.2.3 or how does that get decided who gets which screening tool?
Response Comment 20: Thank you for this important question. We have revised Section 2.2.3 to clarify that all program participants complete a core intake assessment that includes standardized screening tools. However, specific tools—such as for depression or intimate partner violence—are administered based on participant eligibility (e.g., pregnancy/postpartum status), provider discretion, and timing within the perinatal period. This additional detail has been included to clarify screening procedures and improve transparency regarding data collection.
Comment 21 Review 2: Lines 238-240. Save such sentences for discussion section.
Response Comment 21: Thanks for this observation. We have remove interpretive language from the Results section and moving judgmental or explanatory language to the Discussion.
Comment 22 Review 2: Section 3.1 needs a demographics table to be in table 1. Demonstrate who the study population are. Gender by numbers and percentage of total. Median age with IQR, average level of education, etc. Section 3.1 doesn’t even give us the total participants (N) of the study.
Response Comment 22: We appreciate the reviewer’s suggestion. However, we respectfully believe that a separate table is not necessary in this case, as the sociodemographic characteristics of the study population are already clearly described within the narrative of Section 3.1. The information provided includes total participant count, age distribution, gender, participant type, educational attainment, and employment status. Given the descriptive and implementation-focused nature of this evaluation, we believe the current format provides sufficient detail without additional tabular presentation. We added a clear statement for sample size and Section 3.1 has also been updated to reference the sample size.
Comment 23 Review 2: Line 269. What age is “younger age” of pregnancy?
Response Comment 23: Thanks. Corrected.
Comment 24 Review 2: Line 289. Who gets referred? What is the threshold?
Response Comment 24: Thank you for your question. Depression screening is conducted using the Edinburgh Postnatal Depression Scale (EPDS). Participants who score 10 or higher on the EPDS are considered to have clinically relevant depressive symptoms and are referred to mental health services for further evaluation and support. In addition, any participant who responds positively to the self-harm item (item 10) is flagged for immediate referral regardless of total score. These thresholds align with evidence-based guidelines and were applied consistently across all screening encounters. We added the follwoing statement "All screening procedures adhered to the recommended clinical guidelines and scoring thresholds established for each validated tool".
Comment 25 Review 2: Table 1. Delete the whole row of “n” and change every cell to be n/N (%) format showing the number of over the total and percentage. Hence, the ratio of the total number receiving depression screening over the total number of participants from each year and in parenthesis show the percentage.
Response Comment 25: Thanks. Corrected following recommendations.
Comment 26 Review 2: Table 1. Last row should be a test of trend variable with p-value if applicable. Caption of the table should describe which spefici test for trend was used.
Response Comment 26: We appreciate the reviewer’s recommendation. However, the presented analysis is descriptive in nature and not inferential. The data shown in the table reflect raw trends in program outcomes over time and were not subjected to statistical testing for trend, as the intent was to report implementation-level results from a real-world service delivery program. As such, a p-value does not apply to this type of evaluation. We have updated the table caption to clarify that the trends are descriptive and not based on inferential statistical testing.
Comment 27 Review 2: Table 1. “Baseline” is not well described. The caption says 2019-2024. If so, that is not a baseline, that is just an average perhaps. If it is 2019 (year before network of care model fully underway) then please fix the caption. But what is meant by ”baseline” remains unclear.
Response Comment 27: Thank you for pointing out this important clarification. In our analysis, we define the program baseline as calendar year 2019, which reflects the start of participant enrollment prior to the full implementation of the Cuidado Compartido model under Comienzo Saludable Puerto Rico. This year serves as the initial reference point for comparing subsequent changes in outcome indicators. We have revised all tables caption accordingly to ensure that the meaning of "baseline" is clearly defined.
Comment 28 Review 2: Table 1. Many cells unclear. Why is baseline depression screening 100 but then lower after 2020? Why baseline 0 for interconception period? Why downtrend observed from 66.7% in 2020 to 47.8% in 2024? Please describe negative trends as well.
Response Comment 28: Thank you for this valuable feedback. We recognize that some trends in Table 1, such as the decline in depression screening coverage and the absence of baseline data for certain indicators, were not adequately contextualized. In response, we have revised the Results section to explain that the 100% baseline value for depression screening in 2019 reflects a small pilot cohort where all participants were screened; however, as the program scaled up post-2020, screening coverage percentages reflect a larger and more diverse participant population and the 0% baseline for interconception care reflects the fact that this domain was not yet systematically tracked during the initial program year (2019); formal documentation began in 2020. We have also updated the Results narrative in order to present a more balanced interpretation of the data.
Comment 29 Review 2: Same advice comments 24-26 for table 2.
Response Comment 29: Thanks. Corrected following recommendations.
Comment 30 Review 2: Lines 315-317 (last sentence of second paragraph in section 3.2.3 Health Gaps). This statement should be moved to the discussion. Please only provide objective results in this section.
Response Comment 30: Thank you for this suggestion. We agree that interpretive or evaluative language is appropriate for the discussion section. We have removed the statement from the Results section and integrated it into the Discussion.
Comment 31 Review 2: Line 330. “*This year a three-month gap…” Delete this and just mention it in the discussion.
Response Comment 31: Thank you for this suggestion. We agree that interpretive or evaluative language is appropriate for the discussion section. We have removed the statement from the Results section and integrated it into the Discussion.
Comment 31 Review 2: Table 4 & 5 please combine.
Response Comment 31: Thanks. Corrected.
Comment 32 Review 2: Table 6. The “sample activities” and “reach & impact” for CHW Core Training is the exact same phrase copied and pasted. Please fix and provide specifics like the other cells provide some objective numerical values.
Response Comment 32: Thanks. Corrected.
Comment 33 Review 2: Section 3.4.1. Was a thematic analysis done?
Response Comment 33: Yes, a thematic analysis was conducted to examine participants’ responses to open-ended intake questions. This analysis followed a structured, inductive coding process to identify recurring themes across participant narratives. We have revised Section 3.4.1 to explicitly state that a thematic analysis approach was used and included a brief description of the analytic procedures to clarify the methodology.
Comment 34 Review 2: Table 7 rehashes the same statements right above it in section 3.4.2. Keep one and delete the other. But not both.
Response Comment 34: Thank you for this helpful observation. We agree that presenting both the narrative and the table in their current form is redundant. To streamline the section and maintain clarity, we have removed the Table 7 and retained the narrative format.
Comment 35 Review 2: This paper should not just be aiming to praise the organization’s efforts as enthusiastic as the results should It should provide objective and critical analysis of measureable impacts. Please name spefific outcome effects and specific areas of improvement.
Response Comment 35: We appreciate this important reminder to maintain objectivity and focus on measurable outcomes. In response, we have revised both the Results and Discussion sections to clearly highlight specific data-driven improvements—for example, reductions in low birth weight (from 10.21% to 6.76%), preterm births (from 11.34% to 6.80%), and infant mortality (from 8.89% to 0.00%) among participants from 2019 to 2023. Additionally, we emphasize areas needing improvement, such as the decline in depression screening coverage (from 66.7% in 2020 to 47.8% in 2023) and incomplete postpartum visit rates, which reflect structural barriers like transportation and employment constraints. We have also moderated overly positive language throughout the manuscript and reframed interpretive statements to reflect a balanced and critical evaluation of both the strengths and limitations of the program’s impact.
Comment 36 Review 2: So you increase depression screening but that’s also not the symptom most felt (you stated anxiety was) - so should efforts be placed to improve anxiety screening now?
Response Comment 36: Thank you for this insightful observation. While depression screening has been a primary focus—aligned with national perinatal mental health priorities—our findings did reveal that anxiety symptoms were frequently reported, particularly among non-pregnant caregivers and male participants. Tools like the PHQ-9 and anxiety screening (e.g., GAD-7) were consistently and systematically used and applied based on eligibility criteria. There is no data to add on this aspect.
Comment 37 Review 2: Line 389. Is this due to increased awareness or due to the program now being in full effect to take the time to screen and capture each eligible participant.
Response Comment 37: Thank you for raising this important point. The observed increase in screening volume in 2020 likely reflects a combination of factors. First, the program had fully ramped up its implementation by that time, allowing for more consistent and structured integration of screening tools. Second, heightened public awareness of mental health concerns during the early stages of the COVID-19 pandemic may have contributed to increased receptivity among participants and prioritization by staff. We have clarified this point in the revised manuscript to reflect both programmatic maturation and contextual factors as contributors to the trend.
Comment 38 Review 2: Line 395. What are the increased risk groups you mention? Did you identify?
Response Comment 38: Thank you for this helpful observation. We revised the manuscript to explicitly identify the higher-risk groups referenced. These include adolescents aged 15–19, who represented 5.8% of participants and are widely recognized for increased risk of adverse maternal and newborn outcomes; women over 35 (12.9%), associated with elevated clinical risks due to advanced maternal age; as well as non-pregnant caregivers and male partners, who demonstrated distinct psychosocial needs. The revised text provides greater clarity by naming these subgroups and highlighting their relevance to targeted service strategies.
Comment 39 Review 2: Line 413. You mention age distribution but no age distribution is shared anywhere in the paper. This is also why I mentioned to have a demographics table.
Response Comment 39: Thank you for this observation. While detailed sociodemographic information—including age distribution—is included in the narrative format within Section 3.1, we understand the importance of presenting this data in a more accessible form. However, given the brevity and simplicity of the sociodemographic variables, we believe a separate table may be unnecessary and could introduce redundancy. To address your concern, we have clarified age group percentages and participant types directly within the narrative and ensured this information is clearly referenced when discussing age-related risks and outcomes.
Comment 40 Review 2: Thus to include a demographics table, I suggest combinining tables, and deleting Figure 1.
Response Comment 40: Thanks. Corrected.
Comment 41 Review 2: Lines 460-464. Which of these issues are within the scope of what Comienzo Saludable can address and which are outside the scope? Can you discuss more who and how those would be limitations that can be addressed?
Response Comment 41: hank you for this helpful comment. We agree that clarifying which limitations are within the program’s scope and which are systemic challenges is essential to accurately frame the findings. In response, we have revised the discussion of limitations to distinguish between issues that Comienzo Saludable can directly address—such as participant engagement, screening coverage, and care coordination—and those that fall beyond the program’s immediate control, such as health workforce shortages, infrastructure gaps, and policy constraints.
Comment 42 Review 2: Make the discussion much shorter by removing all the praise and overly positive judgement statements.
Response Comment 42: Thanks. Corrected.
Comment 43 Review 2: Section 4.5. What % of the data is missing? Please provide.
Response Comment 43: Thanks. Corrected.
Comment 44 Review 2: Lines 510-511. “Future efforts should focus on sustainability, policy integration, and strategic expansion” - really? you think that much just from what you hae got here with this data?
Response Comment 44: Thank you for this important observation. We acknowledge that the original phrasing may have implied a broader policy recommendation than the data strictly support. Our intent was to highlight potential areas for future inquiry and program development based on observed improvements within the participant population, rather than to assert definitive conclusions. In response, we have revised the sentence to adopt a more measured tone that reflects the scope and limitations of the implementation data. The updated language now emphasizes the value of further evaluation and cautious exploration of these directions.
Comment 45 Review 2: Line 519. Delete the word “significantly” from improved because no formal statistical test was done to show significance - whether statistical or clinical significance.
Response Comment 45: Thanks. Corrected.